# Evolutionary and structural basis of SLAMF1 utilization in morbilliviruses—Implications for host range and cross-species transmission

Ayumu Hyodo[1☉], Fumio Seki[2☉], Kento Fukuda[3], Kaede Tashiro[1,4], Yuki Kitai[1], Yukiko Akahori[1], Hideko Watabe[1], Hiroshi Katoh[1], Rikuto Osaki[5], Daisuke Takaya[5], Norihito Kawashita[6], Hideo Fukuhara[7], Satoshi Ikegame[8,9], Tomoki Yoshikawa[10], Park Eunsil[11] Shigeru Morikawa[11,12], Ryoji Yamaguchi[13], Benhur Lee[8], Katsumi Maenaka[14], Tsuyoshi Shirai[15], Kaori Fukuzawa[5], Shigenori Tanaka[3], Makoto Takeda[1,16]*

1 Graduate School of Medicine and Faculty of Medicine, The University of Tokyo, Bunkyo-ku, Tokyo, Japan, 2 Department of Virology 3, National Institute of Infectious Diseases, Musashimurayama, Tokyo, Japan, 3 Graduate School of System Informatics, Kobe University, Kobe, Hyogo, Japan, 4 Department of Veterinary Medicine, Nihon University, Fujisawa, Kanagawa, Japan, 5 Graduate School of Pharmaceutical Sciences, Osaka University, Suita, Osaka, Japan, 6 School of Science and Engineering, Kindai University, Higashi-osaka, Osaka, Japan, 7 International Institute for Zoonosis Control, Hokkaido University, Sapporo, Hokkaido, Japan, 8 Department of Microbiology, Icahn School of Medicine at Mount Sinai, New York, New York, United States of America, 9 Graduate School of Medical Sciences, Kyushu University, Fukuoka, Japan, 10 Department of Virology 1, National Institute of Infectious Diseases, Musashimurayama, Tokyo, Japan, 11 Division of Experimental Animal Research, National Institute of Infectious Diseases, Shinjuku-ku, Tokyo, Japan, 12 Faculty of Veterinary Medicine, Okayama University of Science, Imabari, Ehime, Japan, 13 Faculty of Agriculture, University of Miyazaki, Miyazaki, Japan, 14 Faculty of Pharmaceutical Sciences, Hokkaido University, Sapporo, Hokkaido, Japan, 15 Faculty of Bioscience, Nagahama Institute of BioScience and Technology, Nagahama, Shiga, Japan, 16 Pandemic Preparedness, Infection and Advanced Research Center, The University of Tokyo, Bunkyo-ku, Tokyo, Japan

☉ These authors contributed equally to this work.
* mtakeda@m.u-tokyo.ac.jp

## Abstract

Morbilliviruses, including measles virus (MV), canine distemper virus (CDV), peste des petits ruminants virus, and cetacean morbillivirus pose a significant threat to humans and animals. While the host range of morbilliviruses is generally well-defined, cross-species transmission events with significant mortality have also been reported. Their entry into immune cells, the primary targets of morbilliviruses, relies on the signaling lymphocytic activation molecule (SLAM), also known as SLAMF1 or CD150. In this study, we hypothesize that the ability of morbilliviruses to utilize heterologous SLAM receptors stems from evolutionarily conserved structural determinants within the SLAM protein and that minimal genetic changes in the viral receptor-binding H protein can enable adaptation to novel hosts. To test this, we systematically assessed SLAM utilization and adaptation by diverse morbilliviruses. We found that most morbilliviruses efficiently utilize SLAM from multiple host species, including *Myotis* bat SLAM, but not human SLAM. Only MV could efficiently utilize human SLAM. Additionally, unlike other morbilliviruses, MV utilized *Myotis* bat SLAM inefficiently. As an example of morbillivirus adaptation to non-host animal SLAM, we

**Data availability statement:** The authors confirm that all data underlying the findings are fully available without restriction. All relevant data are within the paper and its Supporting Information files.

**Funding:** This work was supported by the Japan Agency for Medical Research and Development (JP243fa627001 and 24wm0325063h0002 to MT) the Japan Society for the Promotion of Science KAKENHI (23K21381 to MT), and NIH R01 AI188431 to BL. The funders had no role in study design, data collection and analysis, decision to publish, or preparation of the manuscript.

**Competing interests:** The authors have declared that no competing interests exist.

conducted an MV adaptation experiment with *Myotis* bat SLAM. We demonstrated that MV readily adapted to utilize *Myotis* bat SLAM by acquiring a single N187Y mutation in its hemagglutinin protein. Notably, hypothetical ancestral SLAMs acted as universal receptors for all morbilliviruses. These results reinforced that morbillivirus receptor usage is primarily supported by evolutionarily conserved structural features of SLAM, highlighting a molecular basis that enables morbilliviruses to rapidly adapt to diverse animal SLAMs.

## Author summary

We studied how viruses in the genus *Morbillivirus*, such as measles virus, canine distemper virus, and rinderpest virus, which are notorious for deadly outbreaks in humans and animals, can cause cross-species infections. The host range of morbilliviruses is significantly influenced by a receptor molecule on cell surfaces known as the signaling lymphocytic activation molecule (SLAM). By examining SLAMs from various animals, including humans, dolphins, dogs, seals, and bats, we observed how these viruses can utilize or adapt to utilize non-host animal SLAMs. We found that, in some cases, slight differences in SLAM may act as initial barriers to cross-species transmission. However, these viruses rapidly overcome such barriers, showing a remarkable ability to adapt to and utilize non-host animal SLAMs. Our research highlights the importance of monitoring these viruses to predict and prevent potential cross-species infections, which is crucial for protecting public health and animal welfare over time.

## Introduction

Morbilliviruses, a genus within the *Paramyxovirus* family, comprise a group of highly pathogenic viruses with significant medical and veterinary relevance. They include pathogens that pose substantial health and economic burdens due to their capacity to cause lethal, widespread outbreaks, leading to high mortality rates and economic losses in livestock industries. Each morbillivirus has a distinct host range: measles virus (MV) infects humans, cetacean morbillivirus (CeMV) targets cetaceans, canine distemper virus (CDV) affects carnivores, phocine distemper virus (PDV) infects seals, rinderpest virus (RPV) historically infected cattle, and peste des petits ruminants virus (PPRV) infects small ruminants such as sheep and goats [1,2]. The recent discovery of novel morbilliviruses (or their genomic components) in bats (*Myotis*, *Phyllostomus*, and *Molossus*) [3–5], swine [6], and cats [7] suggests a broader host spectrum than previously recognized [2].

Morbillivirus infection begins when the viral hemagglutinin (H) protein binds to a receptor on the host cell surface. The signaling lymphocytic activation molecule (SLAM), expressed on immune cells, and nectin-4, expressed on epithelial cells, serve as primary receptors [1,2]. Unlike nectin-4, which is highly conserved across

species, SLAM exhibits substantial sequence variability [1], which may, therefore, contribute significantly to the morbillivirus host range [3,8–14]. On the other hand, cross-species transmission events with significant mortality caused by several morbilliviruses have also been reported. Notably, CDV caused large-scale outbreaks in non-human primates in 2006–2008, raising concerns about the potential for morbilliviruses to affect humans [1,15,16]. CeMV has also repeatedly caused interorder transmission to seals [1,17,18].

The ability of viruses to jump between species, known as cross-species transmission, can lead to emerging infectious diseases. Such events typically occur when a virus from an animal host acquires genetic mutations that enhance its ability to infect and replicate in humans. Adaptation to host receptor molecules is often a key determinant in defining viral host range, as seen in multiple emerging viruses [19–21]. While paramyxoviruses, including morbilliviruses, can replicate efficiently in human or primate cell lines expressing the appropriate receptor [3,22], the extent to which SLAM diversity limits or facilitates cross-species transmission remains unclear.

In this study, we hypothesize that the common use of SLAM by morbilliviruses stems from their ability to recognize its evolutionarily conserved structure and that minimal genetic changes in the viral H protein can enable adaptation to non-host animal SLAMs. To test this, we systematically assessed SLAM utilization by diverse morbilliviruses and conducted adaptation experiments. Furthermore, we employed computational and mutational approaches to dissect the molecular basis of receptor usage and adaptation. We also examined the functionality of reconstructed ancestral SLAMs as morbillivirus receptors.

## Results

### Assessment of SLAM utilization by different morbilliviruses

Morbilliviruses have been identified in diverse host species, including humans, cows, goats, dolphins, bats, dogs, and seals. To evaluate their ability to utilize different SLAM receptors (S1 Fig), we generated Vero cells expressing SLAM from these species [14,23,24] (S2 Fig). The Myotis bat SLAM sequence was used as the representative bat SLAM sequence (S1 Fig) [3]. We selected these animal species because they are known or strongly suspected to be natural hosts of morbilliviruses and, therefore, likely to be highly susceptible to infection. We considered that prioritizing the evaluation of SLAMs from these species is biologically and epidemiologically relevant to assessing the receptor compatibility of morbilliviruses. Since antibodies against non-human SLAMs were unavailable, an N-terminal hemagglutinin (HA) epitope tag was introduced, as previously reported [3,25] (S3 Fig). The engineered cells were infected with various morbilliviruses and cytopathic effects (CPEs), characterized by multinucleated giant cell (syncytia) formation, were observed. Due to regulatory restrictions, infectious RPV and PPRV were excluded from these experiments [26,27]. CeMV (muc strain [23]) and CDV (Ac96I strain [22,28]) induced syncytia in Vero cells expressing dolphin-, bat-, dog-, and seal-SLAMs within 24 hours post-infection (hpi) (S4A Fig). PDV (A982 strain [23]) exhibited a similar pattern but required up to 96 hpi (S4B Fig). MV (IC323-EGFP strain [29]) induced syncytia in dolphin-, dog-, and seal-SLAM cells. In bat-SLAM cells, small syncytia and cell rounding were occasionally observed in certain areas. However, CPEs were extremely limited in extent, and no apparent CPEs were detectable in most microscopic fields. Notably, MV was the only morbillivirus capable of inducing syncytia in human-SLAM-expressing cells (S4 Fig). The newly identified Myotis bat morbillivirus (MBaMV) [3] exhibited a unique specificity: it exclusively utilized bat SLAM and failed to induce syncytia in cells expressing SLAM from other species (S4 Fig).

Plaque assays were performed in SLAM-expressing Vero cells to assess viral infectivity and spread quantitatively. MV efficiently formed plaques in human-, dolphin-, dog-, and seal-SLAM cells but showed limited plaque formation in bat-SLAM cells (Fig 1A). MV produced a higher plaque count in dolphin-, dog-, and seal-SLAM cells than in human-SLAM cells, with the largest plaque size observed in dolphin-SLAM cells (Fig 1A). Similarly, CeMV, CDV, and PDV formed plaques in dolphin-, dog-, seal-, and bat-SLAM cells but not in human-SLAM cells (Fig 1A). In contrast to the broader SLAM utilization seen in these morbilliviruses, MBaMV formed plaques exclusively in bat-SLAM cells (Fig 1A). Despite its restricted host specificity, bat SLAM efficiently supported plaque formation by CeMV, CDV, and PDV (Fig 1A), suggesting that bat SLAM retains the key structural motifs necessary for morbillivirus receptor function.

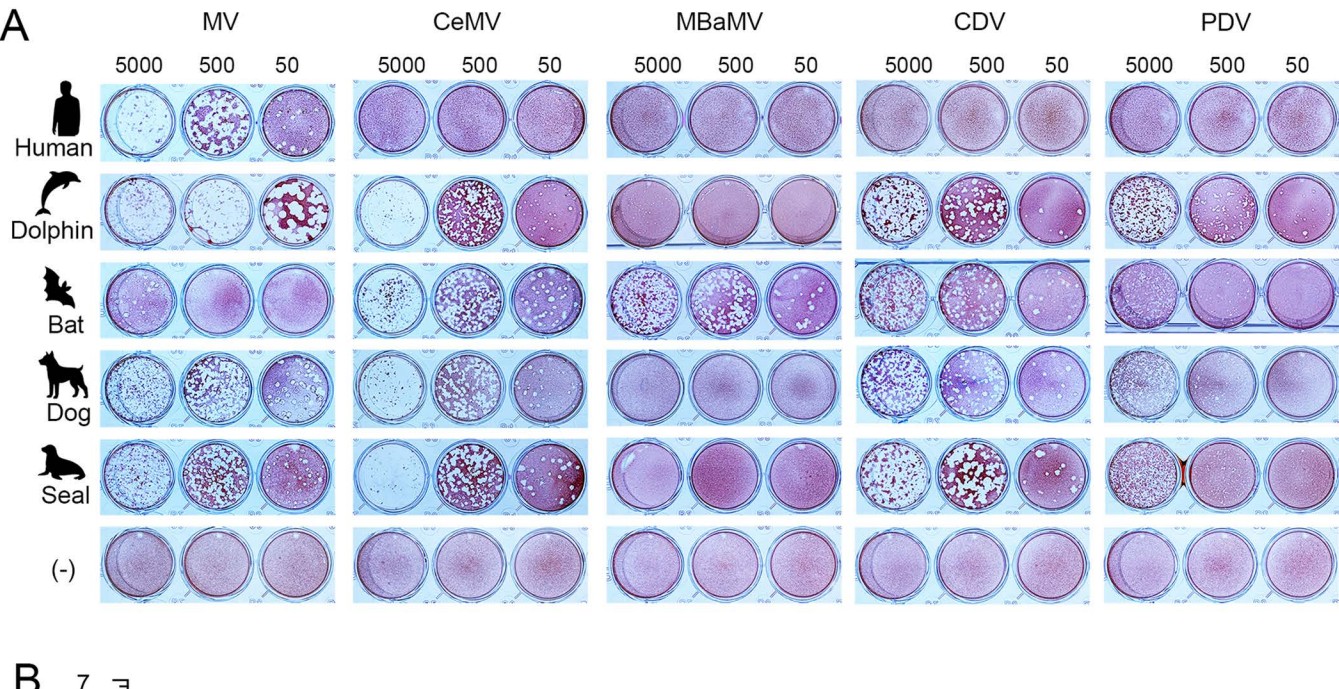

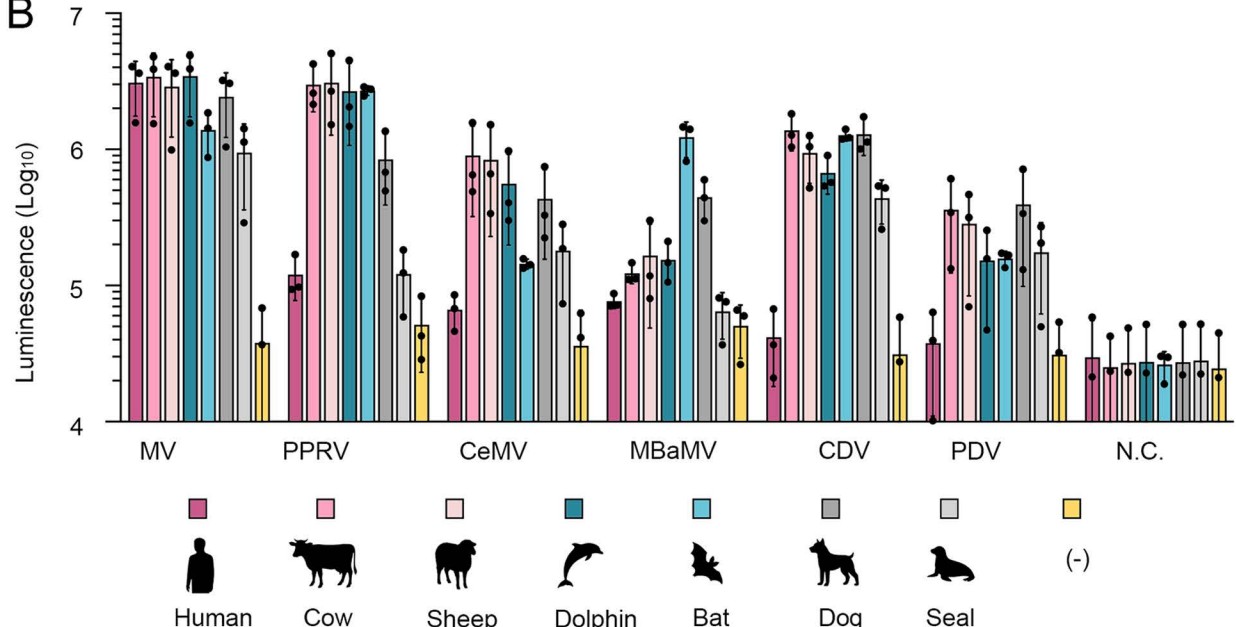

**Fig 1. Plaque formation and quantitative cell fusion assay in various SLAM-expressing cells.** (A) Monolayers of Vero cells stably expressing SLAM from different animal species (human, dolphin, bat, dog, and seal), along with parental Vero cells (-), were infected with different titers (5000, 500, and 50 plaque-forming units [PFUs]) of each morbillivirus (MV, CeMV, MBaMV, CDV, and PDV) and cultured for four days in culture media containing 1% methylcellulose. Cells were stained with neutral red to visualize plaques. The experiment was triplicated, and representative images are shown. (B) **Quantitative cell fusion assay.** 293 CD4/DSP$_{1-7}$ cells transfected with plasmids encoding different animal SLAMs (human, cow, sheep, dolphin, bat, dog, and seal) and 293FT/DSP$_{8-11}$ cells transfected with both H protein- and F protein-encoding plasmids of each morbillivirus (MV, PPRV, CeMV, MBaMV, CDV, and PDV) were mixed at a 1:1 ratio. *Renilla* luciferase activity was measured three days after transfection. 293FT/DSP$_{8-11}$ cells transfected with the MV F protein-encoding plasmid alone served as a negative control (N.C.). Bar chart displaying raw measured values. Means and standard deviations were calculated from triplicate analyses. (A, B) Animal silhouette images were generated using OpenAI's image generation system (DALL·E) and are published under the terms of the Creative Commons Attribution 4.0 International License (CC BY 4.0). For terms of use, see https://openai.com/policies/terms-of-use.

In certain combinations of morbilliviruses and cell lines—particularly with PDV—virus-induced CPE (very small syncytia and cell rounding) was detectable, but plaque formation was barely detectable. To address this, we performed immu-nofluorescence staining at 18 hours post-infection (p.i.), at which only primary infection can be detected (S5 Fig). This approach enabled more sensitive detection of viral infectivity. The results were mostly consistent with those obtained from CPE observations and plaque assays. As expected, however, the differences in infection efficiency dependent on SLAM usage appeared less evident than those observed in the CPE and plaque formation analyses (S5 Fig). This is possibly attributable to two factors: first, CPE and plaque assays reflect multiple rounds of viral infection, thereby amplifying differ-ences in infection efficiency; and second, induction of cell-to-cell fusion requires higher receptor affinity than that needed for virus infection (i.e., virus-to-cell fusion). Indeed, for MV, CeMV, and PDV, infection was consistently observed in Vero cells lacking SLAM expression, at approximately 3–25% of the efficiency in cells expressing the corresponding host animal's SLAM. These findings suggest that while SLAM is essential for cell-to-cell fusion, it is not strictly required for viral entry, but rather serves as a factor that markedly enhances infection efficiency.

## Evaluation of SLAM utilization by morbillivirus H protein in a fusion assay

The infection and plaque assays using infectious viruses and SLAM-expressing cells had two significant limitations. First, the non-human SLAMs expressed in Vero cells contained an N-terminal hemagglutinin (HA) epitope tag, potentially affect-ing receptor function. Second, infectious RPV and PPRV could not be included due to international regulations [26,27]. To overcome these limitations, we conducted a dual-split protein (DSP)-based fusion assay, which enabled us to evaluate SLAM utilization by morbillivirus H and F proteins without using infectious viruses and independently of virus replication capacity. Expression plasmids encoding the H and F proteins from various morbilliviruses, including RPV (KabeteO strain [30]) and PPRV (Ghana/NK1/2010 strain [31]), were paired with different animal SLAMs, including cow and sheep SLAMs. Unlike those in Vero cells, these SLAMs had an authentic, unmodified N-terminus. The results from the DSP fusion assay closely mirrored those from the infection assays, confirming the validity of this approach. Cow and sheep SLAMs func-tioned as receptors for MV, PPRV, CeMV, CDV, and PDV, similar to their respective host SLAMs (Fig 1B). Human SLAM did not support PPRV fusion, reinforcing its restricted usage by non-human morbilliviruses. The DSP assay could not fully assess RPV SLAM usage because the RPV H and F proteins facilitated fusion even without SLAM (S6 Fig). This SLAM-independent fusion may be a laboratory-adapted phenotype of RPV. However, the KabeteO strain is generally con-sidered to retain wild-type characteristics [30].

## Rapid adaptation of MV to utilize bat SLAM

Unlike other morbilliviruses, MV exhibited poor efficiency in utilizing bat SLAM. We conducted an MV adaptation exper-iment with bat SLAM to better understand SLAM recognition by morbilliviruses. After several passages in bat-SLAM cells, MV adapted to replicate and induce syncytia efficiently in bat-SLAM cells. Plaque cloning and subsequent assays demonstrated that the bat SLAM-adapted MV produced clear plaques in bat-SLAM cells, with plaque numbers compara-ble to those in human-SLAM cells (Fig 2A). Additionally, the adapted MV efficiently induced syncytia in bat-SLAM cells—a feature absent in the parental strain but similar to that observed in human-SLAM cells (Fig 2B). These results indicate that MV readily acquired the ability to efficiently utilize bat SLAM while retaining its capacity to use human SLAM. Sequence analysis revealed a single nonsynonymous mutation (N187Y) in the H gene, where asparagine at position 187 was replaced by tyrosine. To assess its functional impact, we introduced the N187Y mutation into an MV H protein expression plasmid, and its effect on cell fusion was examined. The mutation significantly enhanced the fusogenic capacity of the MV H and F proteins in bat-SLAM cells, confirming that this minimal amino acid change enabled efficient bat SLAM utilization (Fig 2C). The analyses up to this point were carried out using bat SLAM with an HA tag (S3 Fig). To eliminate potential effects from the HA tag, Vero cells expressing bat SLAM with an authentic, unmodified N-terminus were generated. Anal-yses using these cells confirmed that MV with the N187Y mutation formed large and numerous syncytia and proliferated

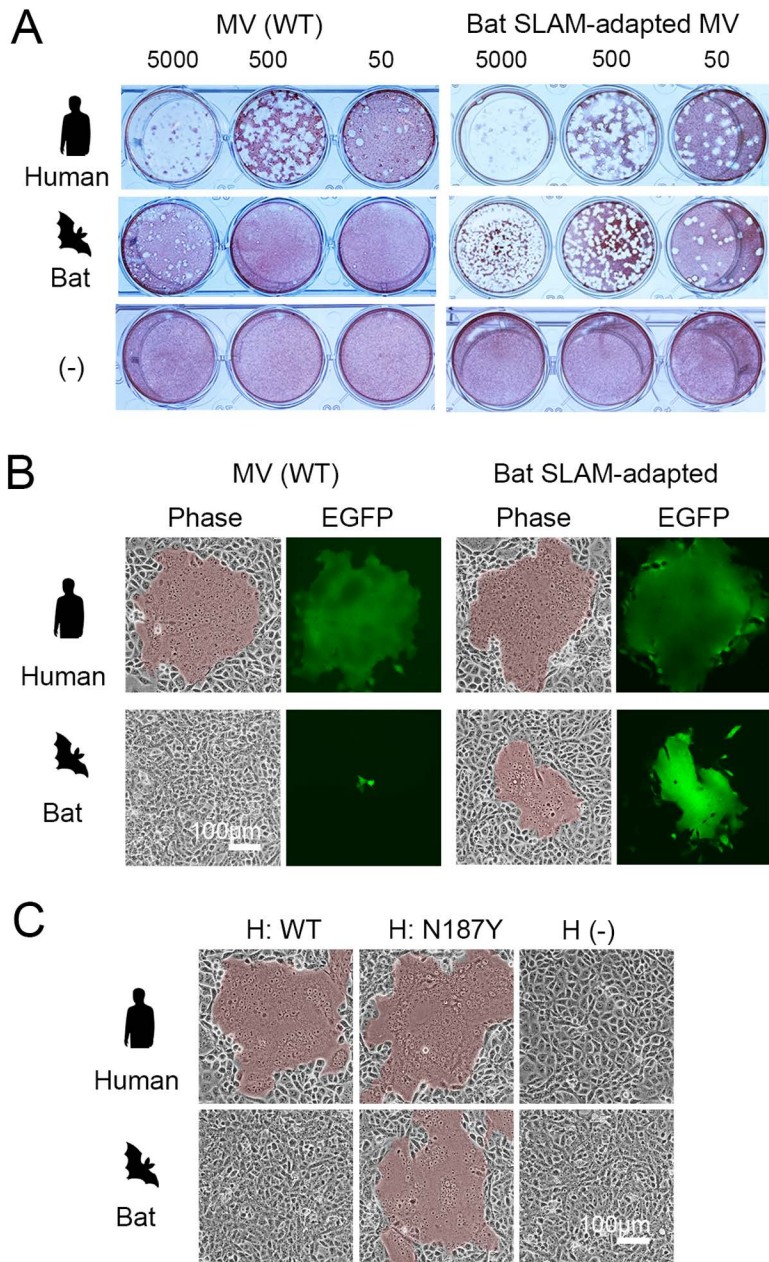

**Fig 2. Characterization of bat SLAM-adapted MV** (A) Plaque formation by wild-type (WT) and bat SLAM-adapted MV in Vero cells expressing human SLAM or bat SLAM, as well as parental Vero cells (-). Plaque assays were performed as described in Fig 1. (B) Syncytium formation by EGFP-expressing wild-type and bat SLAM-adapted MV in Vero cells expressing human SLAM or bat SLAM. (C) Syncytium formation induced by H and F protein expression. The MV F protein was co-expressed with either wild-type MV H protein or MV H protein carrying the N187Y mutation in Vero cells expressing human SLAM or bat SLAM. Cells were observed under a microscope one day post-transfection. Syncytial areas are highlighted in brown. (A–C) Animal silhouette images were generated using OpenAI's image generation system (DALL·E) and are published under the terms of the Creative Commons Attribution 4.0 International License (CC BY 4.0). For terms of use, see https://openai.com/policies/terms-of-use.

in bat-SLAM-expressing cells (S7A Fig). Furthermore, expression plasmid-based analyses reaffirmed that the N187Y mutation was solely responsible for this adaptation (S7B Fig).

## Computational evaluation of SLAM-H interactions

As an example of morbillivirus adaptation to SLAM receptors from different host species, we conducted a detailed computational analysis of MV adaptation to utilize bat SLAM. The first step in the calculation procedure was to create a structure of the complex between the MV H protein and bat SLAM and to perform classical molecular dynamics (MD) calculations on it. A structure was then extracted from the trajectory of the MD calculations, and fragment molecular orbital (FMO) calculations were performed on the extracted structure to evaluate the inter-residue interactions. We can understand why and how particular mutations modify SLAM–H interactions through these computational analyses.

These analyses demonstrated that the N187Y mutation decreased the average inter-fragment interaction energy (IFIE) of the SLAM–H interaction (from −659.5 ± 13.7 kcal/mol to −698.8 ± 14.7 kcal/mol), enhancing attractive interactions. Furthermore, H-Tyr187 formed a new hydrogen bond with a threonine near the N-terminus of bat SLAM (SLAM-Thr25) (Fig 3A and 3B). Additionally, in addition to SLAM-Thr25, H-Tyr187 formed a CH/π interaction with tryptophan at position 112 of SLAM (SLAM-Trp112) (Fig 3C and 3D). In addition, the aspartic acid at position 507 of the H protein (H-Asp507) significantly interacted with the lysine at position 72 of SLAM (SLAM-Lys72) (Fig 3E-3G). In contrast, in MV-H without the N187Y mutation, SLAM-Lys72 interacted mainly with aspartic acid at position 505 (H-Asp505), instead of H-Asp507 (Fig 3E-3G). These two residues (H-Asp505 and H-Asp507) play key roles in MV-H binding to cotton-top tamarin SLAM (PDB ID: 3ALZ) [32] and are highly conserved in morbillivirus H proteins other than the MBaMV H protein (S8 Fig). This shift in interacting residues, induced by the N187Y mutation in the H protein, further supports its role in enhancing the SLAM–H interaction. Analysis of the effect of the N187Y mutation in the MV H protein on its interaction with HA-tagged bat SLAM also revealed molecular interaction changes that enhance the H protein's binding to SLAM (S9 Fig).

Many previous studies have shown that morbilliviruses can change their ability to utilize SLAM through a single amino acid substitution in the H protein. However, detailed molecular analyses of how these point mutations alter the interactions between the H protein and SLAM at the molecular level remain limited. This study presents one such example, demonstrating that a single amino acid change can influence multiple intermolecular interactions. Importantly, this study underscores the potential of computational approaches in elucidating the mechanisms of viral adaptation to new host receptors.

## The receptor function of primate SLAMs for morbilliviruses

Except for MV, all tested morbilliviruses showed limited ability to use human SLAM, suggesting that primate SLAMs may not efficiently function as receptors for these viruses. However, previous studies have demonstrated that CDV strains isolated from monkeys (CYN07 [16] and Monkey-BJ01 [33]) can utilize macaque SLAM. To further investigate the receptor function of primate SLAMs, we analyzed the ability of various CDV strains [28,34–38] to use macaque SLAM and assessed whether this property is restricted to monkey-derived isolates or extends to other CDV strains. Comparative analyses using cells expressing SLAM from macaques, dogs, and humans revealed that most CDV strains isolated from dogs efficiently formed plaques in both macaque- and dog-SLAM cells but not in human-SLAM cells (S10 Fig). Infection titers in macaque-SLAM and dog-SLAM cells were comparable, indicating that CDV can efficiently exploit macaque SLAM.

Notably, among dog CDV strains, the P94S strain [28,34] was uniquely unable to form plaques in macaque-SLAM cells, whereas its closest phylogenetic relative, the Ac96I strain [22], formed plaques efficiently (Fig 4A). Sequence comparison of the H protein between the P94S and Ac96I strains identified a single amino acid substitution (Y539D; tyrosine to aspartic acid at position 539) (GenBank accession numbers AB212964 and AB753775, respectively). To evaluate its functional impact, we introduced the Y539D mutation into the Ac96I H protein expression plasmid and examined its effect in a fusion assay. The mutation did not affect fusion in dog-SLAM cells but impaired it in macaque-SLAM cells (Fig 4B), confirming that Y539D is a key determinant preventing CDV P94S from using macaque SLAM.

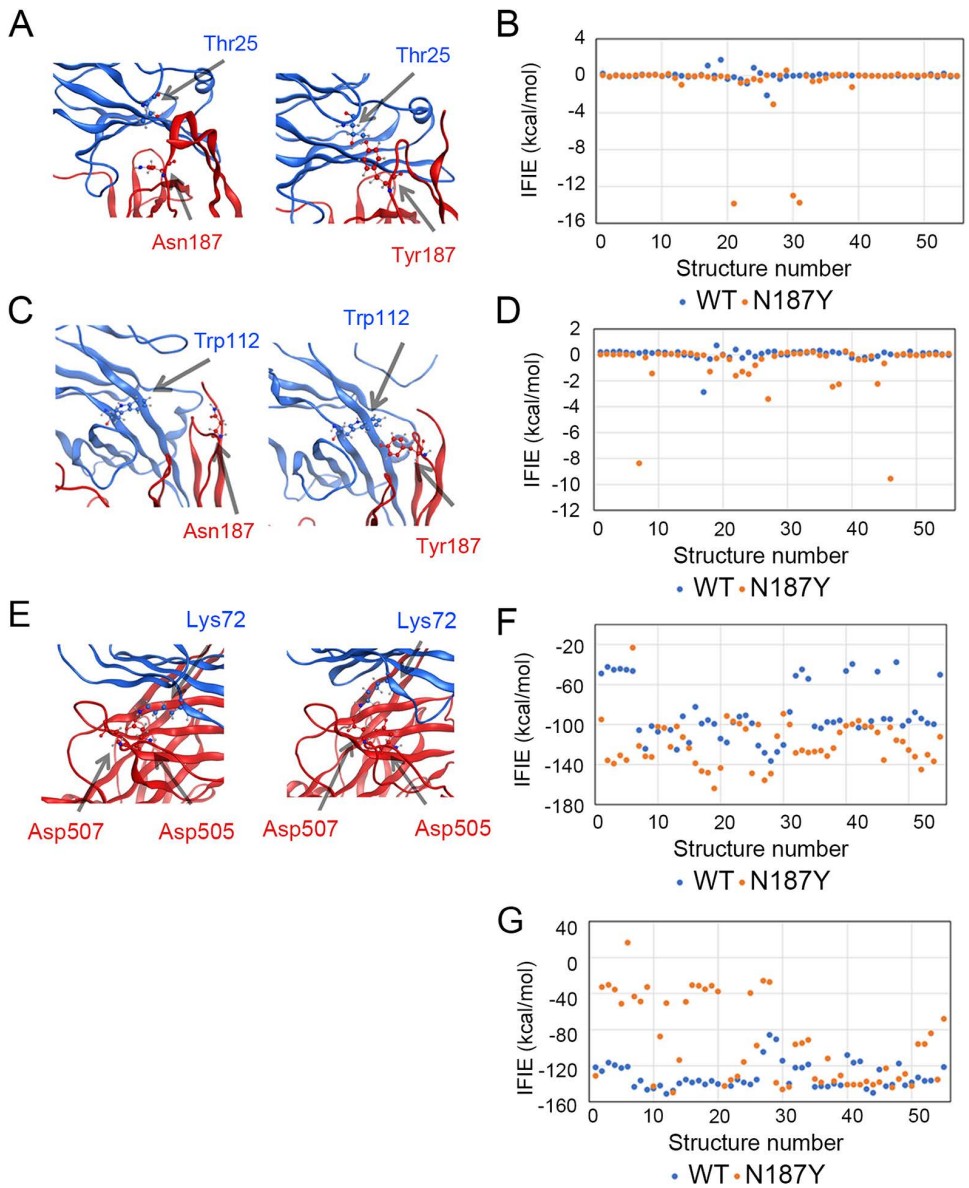

**Fig 3. Computational analysis of inter-residue interactions in the complex of MV-H and bat SLAM.** (A) Positional relationship between H-Asn187 and Thr25 in bat SLAM for the wild type (WT) (left) and between H-Tyr187 and Thr25 in bat SLAM for the N187Y mutant (right). (B) Temporal (structure-dependent) variations of IFIE (kcal/mol) between H-Asn187 (WT) or H-Tyr187 (mutant) and Thr25 in bat SLAM. Blue and orange dots represent WT and the N187Y mutant, respectively. The averaged IFIE values are 0.00 kcal/mol for WT and -0.90 kcal/mol for the mutant. (C) Positional relationship between H-Asn187 and Trp112 in bat SLAM for WT (right) and between H-Tyr187 and Trp112 in bat SLAM for the N187Y mutant (right). (D) Temporal (structure-dependent) variations of IFIE (kcal/mol) between H-Asn187 (WT) or H-Tyr187 (mutant) and Trp112 in bat SLAM. The averaged IFIE values are 0.03 kcal/mol for WT and -0.68 kcal/mol for the mutant. (E) Positional relationship between H-Asp507 or -Asp505 and Lys72 in bat SLAM for WT (left) and the N187Y mutant (right). (F, G) Temporal (structure-dependent) variations of IFIE (kcal/mol) between H-Asp507 and Lys72 in bat SLAM (F) and between H-Asp505 and Lys72 in bat SLAM (G). The averaged IFIE values are -89.12 kcal/mol and -119.23 kcal/mol for H-Asp507–Lys72 in WT and mutant (F), and -132.05 kcal/mol and -95.04 kcal/mol for H-Asp505–Lys72 in WT and mutant (G), respectively.

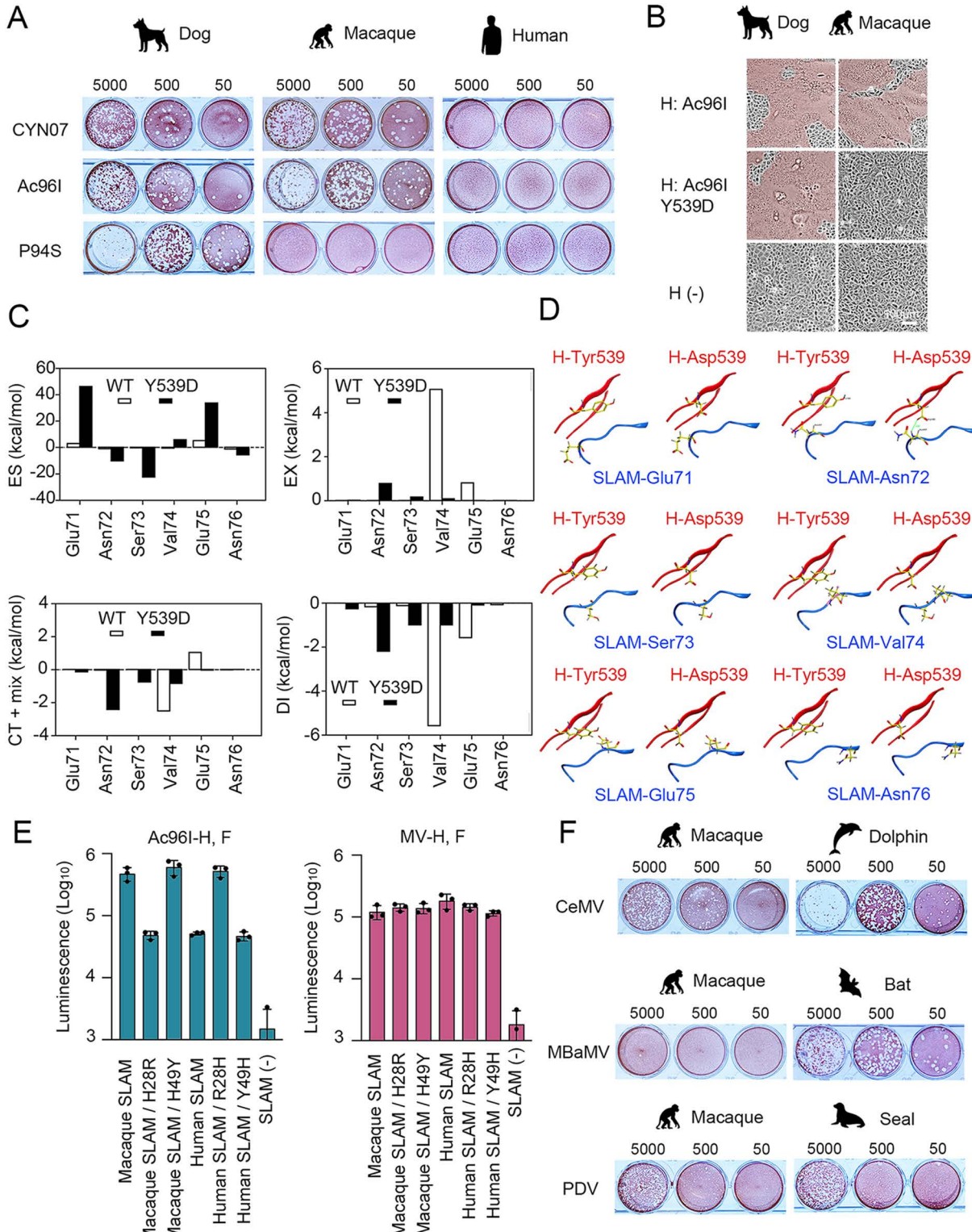

**Fig 4. Receptor function of primate SLAMs as morbillivirus receptors.** (A) Plaque formation by different CDV strains (CYN07, Ac96I, and P94S) in Vero cells expressing dog SLAM, macaque SLAM, or human SLAM. Plaque assays were performed as described in Fig 1. (B) Syncytium formation induced by H and F protein expression. The F protein of the CDV Ac96I strain was co-expressed with the H protein of the same strain, either with or

without the Y539D mutation, in Vero cells expressing dog SLAM or macaque SLAM. Cells were observed under a microscope one day post-transfection. Syncytial areas are highlighted in brown. (C, D) Distribution of interaction energies of each PIEDA term and these interaction pairs in the wild type (FMODBID: L7GV9) and Y539D mutant (FMODBID: 37ZML). (C) Bar graphs show the PIEDA energies (ES, EX, CT+mix, and DI) for interactions between Glu71–Asn76 on SLAM and either Tyr539 or Asp539 on CDV-H. (D) The protein structures, displayed in ribbon representation, illustrate the interaction pairs shown in the bar graphs for the wild type (left) and the Y539D mutant (right). (E) Quantitative fusion assay. Mixed cultures of CHO/DSP$_{1-7}$ and CHO/DSP$_{8-11}$ cells were transfected with plasmids encoding macaque or human SLAM, either with or without mutations at amino acid positions 28 or 49, along with plasmids encoding the H and F proteins of CDV (Ac96I strain) or MV. *Renilla* luciferase activity was measured one day post-transfection. Means and standard deviations were calculated from triplicate analyses. (F) Plaque formation by CeMV, MBaMV, and PDV in Vero cells expressing macaque SLAM or their respective natural host SLAM. Plaque assays were performed as described in Fig 1. (A, B, F) Animal silhouette images were generated using OpenAI's image generation system (DALL·E) and are published under the terms of the Creative Commons Attribution 4.0 International License (CC BY 4.0). For terms of use, see https://openai.com/policies/terms-of-use.

Our group previously modeled the Ac96I-H complex with macaque SLAM [39]. FMO calculations were performed for two model structures—wild-type and mutant CDV-SLAM—to quantitatively assess the interactions of the mutated residue and its surrounding residues. The FMO calculation results for the wild type and Y539D mutant were obtained. Their effects are shown in Fig 4C and 4D. The total IFIE, corresponding to the binding energy between CDV-H and the SLAM receptor, was −869.8 kcal/mol for the wild type and −820.6 kcal/mol for the mutant, indicating that the interaction with the wild type was more stable than Y539D mutant. Changes in Pair Interaction Energy Decomposition Analysis (PIEDA) components between the mutated 539th residue and SLAM residues were observed at SLAM-Glu71, -Asn72, -Ser73, -Val74, -Glu75, and -Asn76. The sum of IFIE for the six residues was 1.7 kcal/mol in the wild type and 39.5 kcal/mol in the mutant, indicating that the mutant is less energetically favorable. The effect of the mutation on each interaction was evaluated in terms of the PIEDA components. In particular, the interaction energies for the pairs involving SLAM-Glu71, -Val74, and -Glu75 were more attractive in the wild type than Y539D. The negatively charged H-Asp539 experienced strong electrostatic repulsion (positive ES energies) from the likewise negatively charged SLAM-Glu71 and SLAM-Glu75, destabilizing the local interaction network. Moreover, the Y539D mutation resulted in the CH/π interaction (DI energy) loss between H-Tyr539 and SLAM-Val74. On the other hand, H-Asp539 formed new hydrogen bonds with SLAM-Ser73 and SLAM-Asn72 and strengthened the electrostatic interaction with SLAM-Asn76, contributing to the stabilization of the mutant. Overall, the significant destabilization of IFIE and PIEDA energies caused by the Y539D mutation suggests repulsion in receptor engagement and is consistent with the virological results showing that the Y539D substitution mutant of the CDV-SLAM complex lost its binding activity. The specific energy values in Fig 4C are shown in S1 Table.

Many CDV strains utilize macaque SLAM but not human SLAM. Previous computational studies suggested that a single amino acid difference at position 28 (His28 in macaque SLAM vs. Arg28 in human SLAM) is critical for this specificity [39] (S1 Fig). To confirm this, a DSP fusion assay was performed. The results showed that human SLAM with histidine at position 28 (His28) functioned as efficiently as macaque SLAM for CDV entry, while macaque SLAM with arginine at position 28 (Arg28) had reduced receptor functionality (Fig 4E). However, it should be noted that although receptor functionality was reduced, a clear distinction remained when compared to SLAM-negative cells, indicating that the receptor was not wholly non-functional. Another mutation at position 49, previously identified between human and macaque SLAMs [39], did not affect receptor function. These findings provide experimental validation of prior *in silico* predictions [39], confirming that the presence of R28 in human SLAM impairs its ability to serve as a functional CDV receptor, although some residual functionality may remain. The receptor function of macaque SLAM for other morbilliviruses (CeMV, MBaMV, and PDV) was also evaluated. CeMV and PDV, but not MBaMV, formed plaques in macaque-SLAM cells at levels comparable to those in their respective host SLAM cells (Fig 4F). This suggests that macaque SLAM can act as a receptor for multiple morbilliviruses, including CDV, CeMV, and PDV.

## The receptor function of hypothetical ancestral SLAMs for morbilliviruses

Our data indicate that SLAMs from *Carnivora* (dogs and seals), *Cetartiodactyla* (dolphins, cows, and sheep), *Chiroptera* (bats), and *Primates* (macaques) function as receptors for various morbilliviruses, despite considerable differences in their amino acid sequences [1]. In this experiment, we also assessed mouse SLAM as a representative of *Rodentia* SLAM to support the phylogenetic reconstruction of ancestral SLAMs and facilitate evolutionary comparisons. We prepared mouse SLAM-expressing Vero cells and showed that CeMV, but not other morbilliviruses, used mouse SLAM as efficiently as the host animal (dolphin) SLAM (S11 Fig). These findings suggest that the species-specific constraints of SLAM as a morbillivirus receptor are more relaxed than previously assumed, allowing for broader cross-species utilization [3,8–14].

To explore the evolutionary changes in SLAM and its role as a morbillivirus receptor, we generated and analyzed hypothetical ancestral SLAMs. We excluded bat SLAM for two main reasons. First, its N-terminal five–amino acid deletion is represented as ambiguous gaps in PAML, which lacks models to accommodate insertions or deletions [40]; this is problematic because the N-terminal region is crucial for receptor usage. Second, including bat SLAM markedly reduces bootstrap support in key phylogenetic branches (S12B Fig), thereby compromising the robustness of the tree topology. Specifically, we reconstructed ancSLAM8, representing the divergence point between *Primates* (humans) and *Rodentia* (mouse), and ancSLAM9, representing the divergence point between *Cetartiodactyla* (cows, sheep, and dolphins) and *Carnivora* (seals and dogs) (Figs 5A and 5B, and S12A). The DSP fusion assay showed that both ancSLAM8 and ancSLAM9 functioned as receptors for all tested morbilliviruses (MV, CeMV, CDV, PDV, MBaMV, and PPRV) (Fig 5C). To validate this with infectious viruses, we generated Vero cells stably expressing ancSLAM8 (S2 Fig). These cells formed syncytia (Fig 5D) and produced plaques with all tested morbilliviruses at levels comparable to their respective host SLAMs (Fig 5E). These results indicate that the hypothetical ancestral SLAMs functioned as broadly permissive receptors for morbilliviruses. Furthermore, our data suggest that fundamental structural properties conserved across SLAMs from different animal orders are important for morbillivirus receptor function.

## Discussion

While our findings emphasize the inherent adaptability of morbilliviruses in utilizing SLAM receptors from diverse species, it is equally important to acknowledge that successful cross-species transmission events—particularly those leading to sustained transmission in a new host population—remain exceedingly rare. This rarity suggests that morbilliviruses may face substantial post-entry barriers in non-native hosts, such as species-specific innate immune responses, intracellular restriction factors, or incompatibilities in replication and transmission dynamics, despite receptor compatibility. Therefore, the observed host restriction is likely multifactorial, and receptor usage alone may not be sufficient to predict zoonotic potential or the likelihood of host-switching events. Nevertheless, we should not be overly optimistic in assuming that, even if non-human morbilliviruses acquire the ability to utilize human SLAM, other host restriction factors will necessarily protect humans from infection. An example of morbillivirus evolution is MV, which is believed to have diverged from an ancestral bovine morbillivirus around 600 BC [41]. Recent studies also suggest human involvement in the emergence and evolution of CDV [42]. Notably, CDV and PPRV can adapt to human SLAM with only one or a few amino acid changes in the H protein [43–45]. However, morbillivirus proteins are antigenically cross-reactive, and immunity against MV can provide some level of cross-protection against infections by different morbilliviruses. However, because exposure to wild-type MV has decreased in recent years, an increasing number of people may lack adequate immunity even to MV. Considering this situation, we should recognize that CDV, which has caused mass mortality in monkeys [1,16,46], poses a considerable risk to humans.

Indeed, adaptation to human receptor molecules is a critical step in the emergence of zoonotic infectious diseases. Various emerging viruses, including coronaviruses and filoviruses, have demonstrated this capacity [19–21]. For paramyxoviruses, including morbilliviruses, receptor specificity is a major determinant of cell tropism and host range [1,15,21,47]. On the other hand, the contributions of other host factors to host-specific barriers remain largely unexplored, apart from

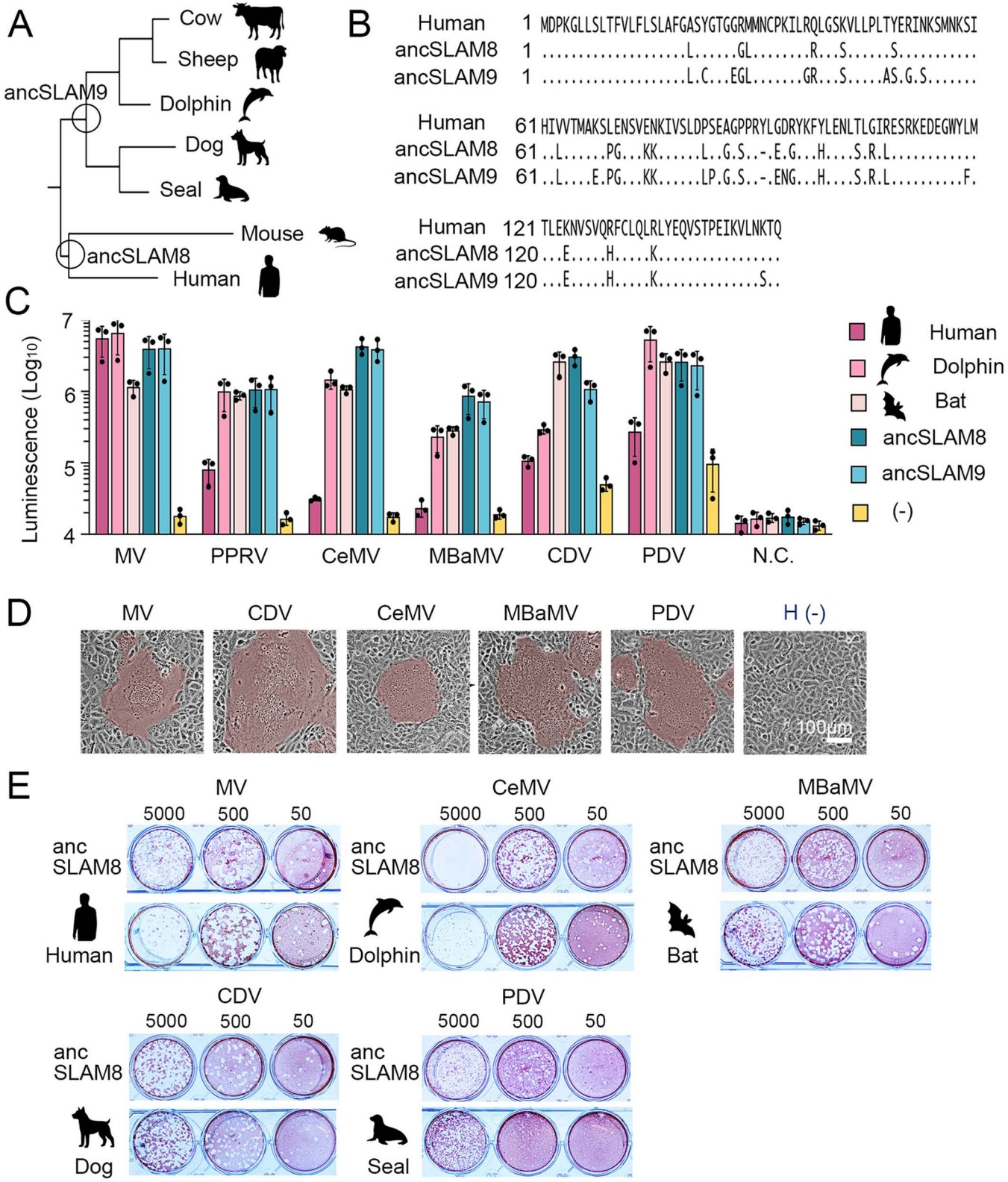

**Fig 5. Characterization of hypothetical ancestral SLAMs.** (A) Phylogenetic tree constructed with animal SLAMs (cow, sheep, dolphin, dog, seal, mouse, and human). The circles indicate the divergence points where the sequence prediction of ancestral SLAM (ancSLAM8 and ancSLAM9)

was performed. (B) Amino acid sequence alignment of the V domains between human SLAM and hypothetical ancestral SLAMs (ancSLAM8 and ancSLAM9). (C) Quantitative cell fusion assay. 293 CD4/DSP$_{1-7}$ cells transfected with plasmids encoding different animal SLAMs (human, dolphin, and bat) as well as ancSLAM8 and ancSLAM9, were mixed with 293FT/DSP$_{8-11}$ cells transfected with plasmids encoding the H and F proteins of each morbillivirus. *Renilla* luciferase activity was measured three days post-transfection. 293FT/DSP$_{8-11}$ cells transfected with the MV F protein-encoding plasmid alone were used as a negative control (N.C.). Bar chart shows raw measured values, with means and standard deviations calculated from triplicate analyses. (D) Syncytium formation induced by H and F protein expression. The H and F proteins of each morbillivirus were expressed in Vero cells expressing ancSLAM8 using expression plasmids. Cells were observed under a microscope one day post-transfection. Syncytial areas are highlighted in brown. (E) Plaque formation by MV, CeMV, MBaMV, CDV, and PDV in Vero cells expressing ancSLAM8 or their respective natural host SLAMs. Plaque assays were performed as described in Fig 1. (A, C, E) Animal silhouette images were generated using OpenAI's image generation system (DALL·E) and are published under the terms of the Creative Commons Attribution 4.0 International License (CC BY 4.0). For terms of use, see https://openai.com/policies/terms-of-use.

limited insights into immune-related barriers [1,15,48]. Non-human paramyxoviruses replicate in human or primate cells when the appropriate receptor is provided [3,22]. This evidence implies that intracellular replication factors may play a less restrictive role than receptor usage.

Although adaptation experiments of animal morbilliviruses to human SLAM or human cells would have provided a direct evaluation of their zoonotic potential, such studies were not conducted in this work due to biosafety and ethical considerations regarding gain-of-function research and dual-use research of concern. Therefore, we focused on experiments with naturally occurring variants and adaptation studies to non-human SLAMs. Although this strategy allowed us to investigate receptor usage and host range within a scientifically rigorous yet ethically responsible framework, this limitation should be acknowledged as a constraint of the present study.

In conclusion, this study shows that evolutionarily conserved structural features primarily govern SLAM's receptor function for morbilliviruses. These features form the basis for the cross-species transmission potential of morbilliviruses. While receptor usage constitutes only one of several barriers to cross-species transmission, this study uncovers SLAM's structural and functional properties and the basis of its adaptability as a morbillivirus receptor. These data are critically important for assessing the zoonotic potential of morbilliviruses and for understanding past and future events of cross-species transmission of these viruses.

## Materials and methods

### Cells

Cell lines such as Vero, Vero/hSLAM (human-SLAM), Vero.DogSLAMtag (dog-SLAMs), Vero/dolphin-SLAM (dolphin-SLAM), and Vero/seal-SLAM (seal-SLAM) were previously reported [14,23,24]. Vero cells expressing *Myotis* bat SLAM and macaque SLAM were previously described [3,16]. However, other clones were newly established for this study. Vero cells were transfected with pCAGGS-Igk-HA-bCD150-P2A-Puro [3] and cultured in the presence of 5 µg/ml puromycin (Nacalai Tesque). Puromycin-resistant Vero cell clones were further incubated with 5 µg/ml puromycin. The expression of bat SLAM (bCD150) was confirmed by flow cytometry using an anti-HA tag monoclonal antibody (clone 16B12, BioLegend, San Diego, CA, USA). The established clone was designated as Vero.BatSLAMtag (bat-SLAM) cells. As no specific antibodies are available for dog, dolphin, seal, and bat SLAMs, these SLAMs were tagged with HA epitope at the N-termini, as previously reported [3,14,23,25]. Vero cells were also transfected with pCXN$_2$-macSLAM [16] and pCXN$_2$-ancSLAM8tag and cultured in the presence of 1.0 mg/ml geneticin (G418; Nacalai Tesque). Geneticin-resistant Vero cell clones were further cultured with 1.0 mg/ml geneticin. Expression of macaque SLAM or ancSLAM8 was verified by flow cytometry using anti-SLAM (clone IPO-3, Kamiya Biomedical) or anti-HA monoclonal (clone 16B12, BioLegend) antibodies (S2 Fig). The clones established were named Vero/macSLAM-6 (macaque-SLAM) and Vero.ancSLAM8tag cells, respectively. Vero cells expressing bat SLAM with the authentic, unmodified N terminus were also generated. The coding region of Igk-HA-bCD150 in the pCAGGS-Igk-HA-bCD150-P2A-Puro plasmid [3] was replaced with that of

bat SLAM with the authentic N terminus, generating pCAGGS-batSLAM-P2A-Puro. Vero cells were transfected with pCAGGS-batSLAM-P2A-Puro and selected in the presence of 10 µg/ml puromycin (Nacalai Tesque). The bulk culture of puromycin-resistant cells, which were predictably expressing bat SLAM, was used for analyses. 293 cells constitutively expressing dual split proteins, namely 293 CD4/DSP$_{1-7}$ and 293FT/DSP$_{8-11}$, were reported previously [49]. CHO cells constitutively expressing dual split proteins were developed in this study. CHO/DSP$_{1-7}$ and CHO/DSP$_{8-11}$ cells were generated by transfecting CHO cells with pRL-DSP$_{1-7}$-neo or pRL-DSP$_{8-11}$-neo, respectively, and selected in DMEM supplemented with 7% FBS and 0.5 mg/ml geneticin. All cell lines were maintained in Dulbecco's Modified Eagle Medium (DMEM) supplemented with 7.5% fetal calf serum (FCS) and antibiotics (100 U/ml penicillin and 0.1 mg/ml streptomycin).

## Viruses

Recombinant wild-type MV expressing enhanced green fluorescent protein (EGFP) has been described previously [29]. Similarly, CeMV muc strain and PDV 982A strain, as well as MBaMV expressing EGFP, have been previously reported [3,23]. The CDV strains isolated using dog-SLAM cells, including Ac96I, S124C, P94S, MD231, MS232, MSA5, Th12, 007Lm, 009L, 011C, M24Cr, and 55L, were also detailed in previous studies [28,34–38,50]. Additionally, the CDV strain isolated from a cynomolgus monkey, designated CYN07-dV, has been previously described [16].

## Plasmid constructions

Mammalian expression plasmids encoding human SLAM (pCA7-hSLAM) and bat SLAM (pCAGGS-Igk-HA-bCD150) were previously described [3,51]. A mammalian expression plasmid encoding bat SLAM with the authentic N terminus was generated by replacing the coding region of Igk-HA-bCD150 in the pCAGGS-Igk-HA-bCD150-P2A-Puro plasmid [3] with that of bat SLAM with the authentic N terminus, generating pCAGGS-batSLAM-P2A-Puro. Mammalian expression plasmids encoding SLAMs for dolphins, dogs, seals, cows, and sheep were constructed by cloning the coding regions of each respective animal SLAM gene into the pCAGGS vector or its derivatives [52]. The DNA fragments encoding these SLAM genes were chemically synthesized based on the sequences available in GenBank, with accession numbers NM_003037.4 (human), XM_004327846.1 (bottlenose dolphin), AF325357 (dog), AB428368 (spotted seal), XP_014402801.1 (riparian myotis bat), BC114833.1 (cow), and DQ228866.1 (sheep). Additionally, DNA fragments encoding hypothetical ancestral SLAM sequences (ancSLAM8 and ancSLAM9) were chemically synthesized, and the cDNA of chimeric SLAM genes with the V domain of ancSLAM8 or ancSLAM9 and the C2 domain of human SLAM were inserted into the pCAGGS vector. The chimeric SLAM cDNA with the V domain of ancSLAM8 and the human SLAM C2 domain was also inserted into the pCXN$_2$ vector with an N-terminal HA-tag, and the constructed plasmid was named pCXN$_2$-ancSLAM8tag.

Additionally, mammalian expression plasmids encoding N-terminally HA-tagged human and macaque SLAM (GenBank accession numbers NM_003037.4 and XM_001117605.3, respectively) were generated using the pCAGGS vector or its derivatives, as previously reported [3,14,23,25]. Site-directed mutagenesis via PCR was employed to introduce amino acid substitutions R28H and Y49H into the HA-tagged human SLAM-expression plasmid, and H28R and H49Y into the HA-tagged macaque SLAM-expression plasmid.

Mammalian expression plasmids for the H and F proteins of MV (wild-type IC-B strain) and the H protein of CDV (Ac96I strain) were previously described [22,53]. Mammalian expression plasmids for the F protein of CDV (Ac96I strain), and the H and F proteins of PPRV, RPV (KabeteO strain), PDV (982A strain), CeMV (muc strain), and MBaMV were constructed by inserting the coding regions of each protein gene into the pCAGGS vector or its derivatives. These viral gene sequences were also synthesized chemically based on GenBank entries, with accession numbers provided in S1 Table. Site-directed mutagenesis was used to introduce mutations; N187Y into the MV H protein-expression plasmid and Y539D into the CDV H protein-expression plasmid.

Expression plasmids for the DSP system (pRL-DSP$_{1-7}$ and pRL-DSP$_{8-11}$) were kindly provided by Dr. Z. Matsuda [54,55]. The region containing the SV40 enhancer/promoter and the coding region of neomycin phosphotransferase was excised from pCI-neo (Promega) and cloned into pRL-DSP$_{1-7}$ and pRL-DSP$_{8-11}$, creating pRL-DSP$_{1-7}$-neo and pRL-DSP$_{8-11}$-neo, respectively.

## Virus infection

Sub-confluent monolayers of cells (specific animal SLAM-expressing Vero cells or their parental Vero cells) were infected with morbilliviruses of interest at a MOI of 0.01, and the cytopathic effect (CPE) in the cells was observed daily Virus infection for 4 days.

## Plaque assay

Confluent monolayers of cells (specific animal SLAM-expressing Vero cells or their parental Vero cells) were incubated with tenfold serial dilutions of virus samples for 1 hour. The cells were then overlaid with DMEM supplemented with 1% methylcellulose and 7.5% fetal calf serum (DMEM/MC/FCS). Four days post-infection, DMEM/MC/FCS containing neutral red was added to the cultures. Plaques stained by neutral red were counted the following day.

## Indirect immunofluorescent assay

Confluent monolayers of cells (specific animal SLAM-expressing Vero cells or their parental Vero cells) were incubated with tenfold serial dilutions of virus samples for 1 hour. At 18 hours p.i., the cells were fixed with PBS containing 10% formalin, and permeabilized with 0.2% Triton X-100. After washing with PBS, the cells were incubated with primary monoclonal antibodies against the MV N protein (clone A56 [56] and E137; a gift from T. A. Sato) for 1 hour, followed by incubation with Alexa Fluor 488-conjugated secondary antibodies (Molecular Probes, Eugene, OR) for 1 hour. Fluorescent signals were then visualized using a fluorescence microscope, and fluorescent infectious foci were counted.

## Conventional cell fusion assay

Sub-confluent monolayers of cells (specific animal SLAM-expressing Vero cells or their parental Vero cells) cultured in 12-well cluster plates were transfected with plasmids encoding the morbillivirus H and F proteins of interest (0.5 μg per well each). Then, the cells were observed for syncytium formation under a microscope daily for 2 days.

## Quantitative cell fusion assay

293FT/DSP8$_{-11}$ cells, cultured in 6-well plates, were transfected with plasmids encoding specific animal SLAM (1.25 μg per well) using the TransIT-293 transfection reagent. Simultaneously, 293 CD4/DSP$_{1-7}$ cells cultured in separate 6-well plates were transfected with plasmids encoding the morbillivirus H and F proteins (2.5 μg and 1.25 μg per well, respectively). Twenty-four hours post-transfection, cells from each culture were resuspended in 3 ml of fresh medium (DMEM/MC/FCS). Then, 92 μl of each resuspended cell suspension was mixed in various combinations and seeded into 96-well plates. After a 15-hour incubation, *Renilla* luciferase activity was quantified using the *Renilla*-Glo Luciferase Assay System (Promega).

For assays using CHO/DSP$_{1-7}$ and CHO/DSP$_{8-11}$ cells, a 1:1 cell mixture was seeded into 24-well plates. These cells were transfected with plasmids encoding the specific animal SLAM receptor (0.125 μg per well) and the morbillivirus H and F proteins of interest (0.125 and 0.25 μg, respectively, per well) using FuGENE HD. *Renilla* luciferase activity was measured the next day using the *Renilla*-Glo Luciferase Assay System (Promega)

## Ancestral SLAM reconstruction

Amino acid sequences of each SLAM were obtained from NCBI with the following accession numbers: human (NP_003028.1), house mouse (AAF22231.1), common bottlenose dolphin (XP_004327894.1), cattle (AAI14834.1), sheep

(ABB58749.1), dog (AAK61857.1), and spotted seal (BAH10672.1). Protein sequences were aligned utilizing Clustal Omega [57]. Using the parameters of Jones-Taylor-Thornton (JTT) model, phylogenetic trees were subsequently generated by the neighbor-joining method with MEGA X [58]. Finally, ancestral SLAM sequences were reconstructed using PAML v4.9j [40]. Alignment gaps were removed from all sequences. The substitution model was based on the empirical JTT matrix, and a uniform substitution pattern was assumed across amino acid sites.

### Computational analysis of the CDV-H Y539D mutation by FMO calculations

The input 3D molecular structures for FMO calculations [59,60] were constructed using MOE [61], employing the wild-type (CDV-H and macaque SLAM complex) model previously developed by our research group [39]. The Y539D mutant structure was constructed from the wild-type model using the Protein Builder module. Generation of hydrogen atom and energy minimization were performed by Structure Preparation module implemented in MOE.

In the energy minimization process, constraint conditions were introduced: atoms within a 4.5 Å radius of the mutated residue (tether = 1.0) and side chain (tether = 0.5). No constrains were introduced for hydrogen atoms. AMBER10:EHT force field was used in all process. FMO calculations were performed on the Fugaku supercomputer, utilizing the ABINIT-MP software (version 1.23) [62]. Fragmentation of proteins was performed in default setting. The IFIE refers to the effective interaction between a pair of specific residues evaluated by the FMO method with inclusion of electron correlation effects by means of Møller-Plesset second-order (MP2) perturbation theory with 6-31G* basis set (FMO-MP2/6-31G*) [63,64]. IFIE and electrostatic interaction (ES), exchange repulsion (EX), charge transfer with higher-order mixed-term interactions (CT+mix) and dispersion interaction (DI) of PIEDA were employed for the evaluation of the interactions. The interaction energy between CDV-H and the SLAM receptor was assessed using IFIE. PIEDA was utilized to analyze the interactions involving Glu71, Asn72, Ser73, Val74, Glu75, and Asn76, which constitute interaction sites between CDV-H and SLAM. The FMO calculation results were deposited in FMODB (https://drugdesign.riken.jp/FMODB/) [65], and the FMODBIDs are L7GV9 (WT) and 37ZML (Y539D).

### Computational analysis of the MV-H N187Y mutation by MD and FMO calculation

The structures of the individual proteins, MV-H (residues 180–610, strain IC-B: NC_001498.1) and bat SLAM (residues 20–139, *Myotis brandtii*: XP_014402801.1) were created using AlphaFold2 [66] based on their respective amino acid sequences. Additionally, the structure of bat SLAM was modeled with an HA tag consisting of 17 residues combined with SLAM residues 27–139. The individual protein structures were then superimposed using MOE [61] onto the complex structure of MV-H and cotton-top tamarin SLAM (PDB ID: 3ALZ) [32], generating three-dimensional models of the MV-H and bat SLAM complex, both with and without the HA tag. The residue at position 187 in MV-H was subsequently mutated from asparagine (N) to tyrosine (Y) (N187Y) using MOE, resulting in four different MV-H and bat SLAM complex structures: with or without the HA tag on bat SLAM and with or without the N187Y mutation in MV-H. Finally, homology modeling was performed on the N-terminal regions of SLAM in the four complex structures (residues 20–31 for the untagged structure, residues 1–17 of the HA tag, and SLAM residue 27 for the tagged structure) to refine any unnatural structural features. In the procedure described above, AlphaFold2 was executed on Google Colab [67] without a template, and the structure with the highest pLDDT (predicted local distance difference test) score was selected from the five generated models. For structural optimization using MOE, the AMBER10:EHT force field was employed.

MD calculations on the modeled complex structures in a periodic simulation cell of solvated water were executed using GROMACS [68] on the Fugaku supercomputer. Structural sampling was performed to extract snapshots from the simulation trajectory. TIP3P [69] was used as the water force field, and ff14SB [70] was employed as the protein force field. The calculation protocol consisted of structure optimization, heating process, density relaxation, equilibration process, and main calculation, in that order. Temperature and pressure were controlled using the Berendsen thermostat

and the C-rescaling method, respectively. Five independent 100 ns MD calculations were performed for structures containing tagged SLAM. Eleven snapshots were extracted from each trajectory at 10 ns intervals, yielding 55 structures. Large fluctuations, primarily associated with SLAM-Asp20, were observed for structures containing untagged SLAM. Therefore, ten independent 100 ns MD calculations were performed to eliminate outlier structures exhibiting such fluctuations. Eleven snapshots were extracted from each trajectory at 10 ns intervals, yielding 110 structures. These structures were then subjected to structural filtering. First, structures in which the nearest neighbor distance exceeded 2.0 Å were excluded. This criterion was applied to the distance between SLAM-Lys72 and its strongest interacting residue, H-Asp505 in the wild type and H-Asp507 in the N187Y mutant, and the distance between SLAM-Glu119 and H-Arg533. Subsequently, 55 structures were selected based on the increasing distance between SLAM-Asp20 and the nearest residue in the MV-H protein.

FMO calculations were conducted on the extracted structure to assess inter-residue interactions. For each of the 55 structures thus obtained and locally optimized by MOE, FMO-MP2/6-31G* calculations were performed in the same way as for the CDV-H Y539D mutation. The overall interaction energy between the MV-H and SLAM proteins was calculated by summing the IFIE values. The FMO calculation results will be deposited in FMODB (https://drugdesign.riken.jp/FMODB/) [65].

## Supporting information

**S1 Table.** IFIE and PIEDA energy components between 359th residue of MV-H and each residue of SLAM. (DOCX)

**S2 Table.** GenBank accession numbers for morbillivirus H/RBP and F genes. (DOCX)

**S1 Fig. Multiple alignment of the V domains of various animal SLAMs.** Amino acid sequences of SLAMs from various animals were obtained from GenBank and aligned using the ClustalW program in Genetyx-Mac software ver. 21. Residues identical to the top sequence are represented by dots, while mismatched residues are shown as their respective amino acid letters. Conserved residues across all sequences are highlighted in black, whereas residues conserved in more than half of the sequences are highlighted in gray. Deleted amino acid residues are indicated by hyphens. The amino acid positions shown to affect the H-SLAM interaction in this study are indicated by red dots. (TIF)

**S2 Fig. Establishment of Vero cell lines stably expressing different SLAMs.** Vero.BatSLAMtag (filled green profile, left panel), Vero.ancSLAM8tag (filled green profile, right panel), and parental Vero cells (filled gray profile, left and right panels) were stained with a mouse anti-HA tag monoclonal antibody (clone 16B12, BioLegend), followed by Alexa Fluor 488-conjugated anti-mouse IgG staining. Vero/macSLAM-6 (filled green profile, center panel) and parental Vero cells (filled gray profile, center panel) were stained with a mouse anti-SLAM monoclonal antibody (clone IPO-3, Kamiya Biomedical), followed by Alexa Fluor 488-conjugated anti-mouse IgG staining. (TIF)

**S3 Fig. N-terminal Sequence of animal SLAMs with an HA tag.** The HA tag sequence, SLAM sequence, and linker connecting the HA tag and SLAM sequence are highlighted in purple, blue, and orange, respectively. Residues identical to the top sequence are represented as dots, while mismatched residues are shown as their corresponding amino acid letters. Deleted amino acid residues are indicated by hyphens in the alignment analysis. Accession numbers: DQ228866.1 (sheep), BC114833.1 (cow), XM_004327846.1 (bottlenose dolphin), AF325357 (dog), AB428368 (spotted seal), NM_003037.4 (human), and XP_014402801.1 (riparian myotis bat). and XM_001117605.3 (macaque). (TIF)

PLOS Pathogens

**S4 Fig. Cytopathic effects induced by morbilliviruses in various SLAM-expressing cells.** Vero cells stably expressing SLAM from different animal species (human, dolphin, bat, dog, and seal), along with parental Vero cells (denoted as '-'), were infected or mock-infected with measles virus (MV), cetacean morbillivirus (CeMV), myotis bat morbillivirus (MBaMV), canine distemper virus (CDV), or phocine distemper virus (PDV) at a multiplicity of infection (MOI) of 0.01. Cytopathic effects (CPEs) were evaluated at 24-hour intervals. (A) CPE observations at 24 hours post-infection (hpi). (B) CPE observations at 96 hours post-infection. CPE scoring: 2 + : Large syncytia observed throughout the field of view; + : Few small syncytia detected; (-): No syncytia detected; >2 + : Majority of cells detached (no image shown). Syncytial areas are highlighted in brown. Animal silhouette images were generated using OpenAI's image generation system (DALL·E) and are published under the terms of the Creative Commons Attribution 4.0 International License (CC BY 4.0). For terms of use, see https://openai.com/policies/terms-of-use.
(TIF)

**S5 Fig. Infectivity assay of morbilliviruses in various SLAM-expressing cells.** Vero cells stably expressing SLAM from different animal species (human, dolphin, bat, dog, and seal), along with parental Vero cells (denoted as '–'), were incubated with tenfold serial dilutions of virus samples (MV, CeMV, MBaMV, CDV, or PDV) for 1 hour. At 18 hours p.i., the cells were fixed with PBS containing 10% formalin, and permeabilized with 0.2% Triton X-100. Infected cells were detected using primary monoclonal antibodies against the MV N protein (clones A56 and E137), followed by Alexa Fluor 488-conjugated secondary antibodies. Fluorescent signals were visualized using a fluorescence microscope, and fluorescent infectious foci were counted. Animal silhouette images were generated using OpenAI's image generation system (DALL·E) and are published under the terms of the Creative Commons Attribution 4.0 International License (CC BY 4.0). For terms of use, see https://openai.com/policies/terms-of-use.
(TIF)

**S6 Fig. Syncytium Formation by rinderpest virus (RPV) H and F proteins in the absence of SLAM expression.** The RPV H and F proteins were expressed in HeLa cells using expression plasmids. As a control, the RPV F protein was also expressed alone in HeLa cells. One day post-transfection, the cells were observed under a microscope.
(TIF)

**S7 Fig. Characterization of bat SLAM-adapted MV in Vero cells expressing bat SLAM with an authentic, unmodified N terminus. (A)** Syncytium formation by EGFP-expressing MBaMV, wild-type MV, and bat SLAM-adapted MV. (B) Syncytium formation by the H protein and F protein-encoding plasmids. The H and F proteins of MBaMV or MV were expressed in Vero cells expressing bat SLAM with an authentic, unmodified N terminus using expression plasmids. The MV H protein carrying the N187Y mutation was also expressed in these cells. One day post-transfection, the cells were observed under a microscope.
(TIF)

**S8 Fig. Multiple alignment of morbillivirus H proteins.** Amino acid sequences of H proteins from seven morbilliviruses were obtained from GenBank and aligned using the ClustalW program in Genetyx-Mac software ver. 21. Residues identical to the top sequence are represented by dots in red boxes, while mismatched residues are shown as their respective amino acid letters. Deleted amino acid residues are indicated by hyphens. The amino acid positions shown to affect the H-SLAM interaction in this study are indicated by blue dots.
(TIF)

**S9 Fig. Computational analysis of inter-residue interactions in the complex of MV-H and HA-tagged bat SLAM.** (A) Amino acid sequences of bat and human SLAMs in the absence (upper panel) and presence (lower panel) of a tag and linker. Predicted signal peptide sequence, HA tag sequence, SLAM sequence, and linker connecting the HA tag and

SLAM sequence are highlighted in green, purple, blue, and orange, respectively. Residues identical to the top sequence are represented as dots, while mismatched residues are shown as their corresponding amino acid letters. Deleted amino acid residues are indicated by hyphens in the alignment analysis. (B) MD trajectory-averaged total IFIE (kcal/mol) between MV-H and bat SLAM with a tag. Blue and orange bars indicate the FMO-calculated results, showing the mean values and standard deviations, for the wild type (WT) and N187Y mutant, respectively. (C) Positional relationship between H-Asn187 and tag15 (Arg) in bat SLAM with a tag for the wild type (WT) (left) and between H-Tyr187 and tag15 (Arg) for the N187Y mutant (right). (D) Temporal (structure-dependent) variations of IFIE (kcal/mol) between H-Asn187 or H-Tyr187 and tag15 (Arg24) in bat SLAM with a tag. Blue and orange dots represent the wild type (WT) and the N187Y mutant, respectively. The averaged IFIE values are -1.3 kcal/mol for WT and -5.10 kcal/mol for the mutant. (E) Positional relationship between H-Arg610 and tag7 (Asp) in bat SLAM with a tag for WT (left) and for the N187Y mutant (right). (F) Temporal (structure-dependent) variations of IFIE (kcal/mol) between H-Arg610 and tag7 (Asp16) in bat SLAM with a tag. The averaged IFIE values are -18.4 kcal/mol for WT and -78.6 kcal/mol for the mutant. (G) Positional relationship between H-Asp507 or -Asp505 and Lys72 in bat SLAM with a tag for WT (left) and the N187Y mutant (right). (H, I) Temporal (structure-dependent) variations of IFIE (kcal/mol) between H-Asp507 and Lys72 in bat SLAM with a tag (H) and between H-Asp505 and Lys72 (I). The averaged IFIE values are -78.8 kcal/mol and -103.9 kcal/mol for H-Asp507–Lys72 in WT and mutant (H), and -109.93 kcal/mol and -76.6 kcal/mol for H-Asp505–Lys72 in WT and mutant (I), respectively.
(TIF)

**S10 Fig. Plaque formation by various CDV strains.** Monolayers of Vero cells stably expressing dog, macaque, and human SLAMs in 12-well cluster plates were infected with different infectious titers (5,000, 500, and 50 plaque-forming units [PFUs]) of each CDV strain and cultured for four days in 1% methylcellulose-containing culture medium. The cells were stained with neutral red to visualize the plaques. Animal silhouette images were generated using OpenAI's image generation system (DALL·E) and are published under the terms of the Creative Commons Attribution 4.0 International License (CC BY 4.0). For terms of use, see https://openai.com/policies/terms-of-use.
(TIF)

**S11 Fig. Plaque formation by morbilliviruses in Vero cells expressing mouse SLAM or their respective natural host SLAM.** (A) Monolayers of Vero cells stably expressing SLAM from mouse or respective natural host SLAM, along with parental Vero cells (-), were infected with different titers (5000, 500, and 50 plaque-forming units [PFUs]) of each morbillivirus (MV, CeMV, MBaMV, CDV, and PDV) and cultured for four days in culture media containing 1% methylcellulose. Cells were stained with neutral red to visualize plaques. The experiment was triplicated, and representative images are shown. Animal silhouette images were generated using OpenAI's image generation system (DALL·E) and are published under the terms of the Creative Commons Attribution 4.0 International License (CC BY 4.0). For terms of use, see https://openai.com/policies/terms-of-use.
(TIF)

**S12 Fig. Phylogenetic analysis of animal SLAMs with and without bat SLAM.** (A) Detailed phylogenetic tree corresponding to Fig 5A. Each branch is labeled with its bootstrap probability based on 1,000 replicates. (B) Phylogenetic tree of animal SLAMs that includes bat, cow, sheep, dolphin, dog, seal, and human. Sequences were aligned with Clustal Omega. The topology was inferred via the neighbor-joining method using the Jones-Taylor-Thornton (JTT) substitution model. (C) Amino acid sequence alignment of the V domains from animal SLAMs (human, mouse, cow, sheep, dolphin, dog, and bat), ancSLAM8, ancSLAM9, and the newly inferred ancestral SLAMs. These ancestral sequences were reconstructed using PAML v4.9j (with the pairwise deletion option) based on an input dataset that included bat SLAM, and were subsequently aligned with Clustal Omega.
(TIF)

**S1 Data.** The complete set of raw data used to generate the graphs in **Figs 1B**, **4E**, **5C**, and **S5**. (XLSX)

## Acknowledgments

We thank Drs. Zene Matsuda and Mizuki Yamamoto for providing the DSP system, and Dr. Yusuke Yanagi for providing various reagents related to SLAM and recombinant MVs. We also thank Drs. Tadashi Maruyama, Yoshiharu Mori, Kazue Ohishi, Hiroaki Tokiwa, and Yuta Yamamoto for their invaluable support and suggestions. We also sincerely thank Kyoritsu Seiyaku Co., Ltd. for providing CDV strains for this study. This study was performed as part of the activities of the FMO Drug Design Consortium (FMODD) using the Fugaku supercomputer (project ID: hp240162).

## Author contributions

**Conceptualization:** Ayumu Hyodo, Shigenori Tanaka, Makoto Takeda.

**Data curation:** Rikuto Osaki, Kaori Fukuzawa, Shigenori Tanaka.

**Formal analysis:** Rikuto Osaki, Kaori Fukuzawa, Shigenori Tanaka.

**Funding acquisition:** Hiroshi Katoh, Benhur Lee, Makoto Takeda.

**Investigation:** Ayumu Hyodo, Fumio Seki, Kento Fukuda, Kaede Tashiro, Yuki Kitai, Yukiko Akahori, Hideko Watabe, Hiroshi Katoh, Rikuto Osaki, Daisuke Takaya, Norihito Kawashita, Hideo Fukuhara, Tomoki Yoshikawa, Park Eunsil, Katsumi Maenaka, Shigenori Tanaka, Makoto Takeda.

**Methodology:** Ayumu Hyodo, Fumio Seki, Tsuyoshi Shirai, Kaori Fukuzawa, Shigenori Tanaka, Makoto Takeda.

**Project administration:** Hiroshi Katoh, Shigenori Tanaka, Makoto Takeda.

**Resources:** Fumio Seki, Hiroshi Katoh, Satoshi Ikegame, Shigeru Morikawa, Ryoji Yamaguchi, Benhur Lee, Kaori Fukuzawa, Shigenori Tanaka, Makoto Takeda.

**Software:** Kaori Fukuzawa, Shigenori Tanaka.

**Supervision:** Shigenori Tanaka, Makoto Takeda.

**Validation:** Ayumu Hyodo, Fumio Seki, Kento Fukuda, Tsuyoshi Shirai, Shigenori Tanaka, Makoto Takeda.

**Writing – original draft:** Ayumu Hyodo, Shigenori Tanaka, Makoto Takeda.

**Writing – review & editing:** Ayumu Hyodo, Shigenori Tanaka, Makoto Takeda.

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
