## [Decision Letter · Decision Letter 0]

PPATHOGENS-D-25-00434

Evolutionary and structural basis of SLAM utilization in morbilliviruses – Its implications for host range and cross-species transmission

PLOS Pathogens

Dear Dr. Takeda,

Thank you for submitting your manuscript to PLOS Pathogens. After careful consideration, we feel that it has merit but does not fully meet PLOS Pathogens's publication criteria as it currently stands. Therefore, we invite you to submit a revised version of the manuscript that addresses the points raised during the review process. Your manuscript has now been carefully evaluated by expert reviewers in the field. As you can see from their comments, all reviewers felt that your study was of interest, but they each raised several important points that require your attention. While the majority of the points raised by each reviewer can likely be addressed through revisions/clarifications to the text, several of the points raised by reviewers may require the inclusion of additional experimental data. Overall, but specifically for the abstract, a clear hypothesis and goal should be provided. The reviewers bring up a point that when addressing the zoonotic potential for morbilliviruses, the fact that no other morbillivirus uses the human receptor argues against this. While trying to understand what is need for measles virus to infect bats (reverse zoonoses?) is scientifically interesting, a more relevant question would be, what is needed to be able to use a human receptor as suggested by reviewers 1 and 2. In addition, while recognizing the restraints of using some of these viruses, the fusion assay does not necessarily reflect infection. This should be, at least for some viruses, confirmed using other methods as suggested by reviewer 3. If you are able to fully address my concerns and those of the reviewers, we would encourage you to submit a revised manuscript, however, please note a resubmitted manuscript may be subjected to full editorial and peer review.

Please submit your revised manuscript within 60 days May 10 2025 11:59PM. If you will need more time than this to complete your revisions, please reply to this message or contact the journal office at plospathogens@plos.org. Please include the following items when submitting your revised manuscript:

We look forward to receiving your revised manuscript.

Kind regards,

Barry Rockx

Academic Editor

PLOS Pathogens

Thomas Hoenen

Section Editor

PLOS Pathogens

 Sumita Bhaduri-McIntosh

Editor-in-Chief

PLOS Pathogens

orcid.org/0000-0003-2946-9497 Michael Malim

Editor-in-Chief

PLOS Pathogens

orcid.org/0000-0002-7699-2064

**Journal Requirements:**

At this stage, the following Authors/Authors require contributions: Ayumu Hyodo, Fumio Seki, Kento Fukuda, Kaede Tashiro, Yuki Kitai, Yukiko Akahori, Hideko Watabe, Hiroshi Katoh, Rikuto Osaki, Daisuke Takaya, Norihito Kawashita, Hideo Fukuhara, Satoshi Ikegame, Tomoki Yoshikawa, Park Eunsil, Shigeru Morikawa, Ryoji Yamaguchi, Benhur Lee, Katsumi Maenaka, Tsuyoshi Shirai, Kaori Fukuzawa, Shigenori Tanaka, and Makoto Takeda. Please ensure that the full contributions of each author are acknowledged in the "Add/Edit/Remove Authors" section of our submission form.

2) We have noticed that you have uploaded Supporting Information files, but you have not included a list of legends. Please add a full list of legends for your Supporting Information files after the references list.

3) Some material included in your submission may be copyrighted. According to PLOSu2019s copyright policy, authors who use figures or other material (e.g., graphics, clipart, maps) from another author or copyright holder must demonstrate or obtain permission to publish this material under the Creative Commons Attribution 4.0 International (CC BY 4.0) License used by PLOS journals. Please closely review the details of PLOSu2019s copyright requirements here: PLOS Licenses and Copyright. If you need to request permissions from a copyright holder, you may use PLOS's Copyright Content Permission form.

Potential Copyright Issues:

- Figures 1, 2, 3, 4, 6, 7, S8, and S9. Please confirm whether you drew the images / clip-art within the figure panels by hand. If you did not draw the images, please provide a link to the source of the images or icons and their license / terms of use; or written permission from the copyright holder to publish the images or icons under our CC BY 4.0 license. Alternatively, you may replace the images with open source alternatives. See these open source resources you may use to replace images / clip-art:

4) Please ensure that the funders and grant numbers match between the Financial Disclosure field and the Funding Information tab in your submission form. Note that the funders must be provided in the same order in both places as well.

**Reviewers' Comments:**

Reviewer's Responses to Questions

**Part I - Summary**

Reviewer #1: In this manuscript, Hyodo et al evaluate whether SLAMs from different species support morbillivirus entry and subsequent replication. Additionally, they perform adaptation experiments of MV to batSLAM, and explore why CDV can use macaqueSLAM but not humanSLAM. The manuscript has interesting findings that are worth publishing, but the data is presented in a rather chaotic form and hard to follow for a general readership.

Reviewer #2: The authors have studied the interactions of a range of morbilliviruses with CD150 of different host species, using the combination of infection and fusion assays. This study is partially built on previous work of the same authors (Takeda et al., Curr Opin Virol 2020, especially table 1 and figure 2). They identified situations where single point mutations explained if a specific CD150 molecule could be used as cellular receptor. In addition, they showed that in vitro adaptation of viruses to use a novel receptor was straightforward. In addition, the authors constructed hypothetical ancestral CD150 molecules that may have acted as universal morbillivirus receptors.

Reviewer #3: The study examines the usage of SLAM receptors from different species by measles and the veterinary morbilliviruses. The manuscript is well written and the experiments valid and precisely carried out. However, the evidence of virus growth in the cell lines is primarily based on cell fusion. There is no immunofluorescence data. This is particulalry important for PDV, where some wild type strains have been shown to cause cell rounding rather than fusion and to grow to good titres in Vero cells without a SLAM receptor (this has been how in a number of laboratories). The authors need to carry out immunofluorescense staining to confirm their conclusions.

A minor point is that seals are carnivores.

**Part II – Major Issues: Key Experiments Required for Acceptance**

Reviewer #1: Major comments

- The authors continuously use different combinations of SLAMs and morbilliviruses, without providing a logical rationale for that. Actually, the authors don’t present a clear hypothesis and goal in this manuscript at all. This makes the manuscript chaotic and difficult to read. A clear hypothesis and goal should be added to the abstract and end of introduction.

- The authors show that many animal morbilliviruses can use SLAMs from different host species, concluding that morbilliviruses are adaptable. However, human SLAM is inaccessible to any other morbillivirus than MV. So, if the authors want to make a case that animal morbilliviruses could jump to humans (line 84), why was this adaptation to human not investigated? Rather the authors investigated the adaptability of MV to batSLAM and CDV to macaqueSLAM, which seems completely irrelevant? Why was macaque SLAM not included in the first 2/3 figures?

- Why was mouse SLAM suddenly included in the experiments with the ancestral SLAM. Why was it not included in earlier datasets (first 2/3 figures).

- Splitting the abstract into two pieces, with a second part specifically on bats, does not help readability. The specific focus on the bat results in the abstract, to a detailed level, is unnecessary. The bat data should be integrated, and a single cohesive abstract should be presented.

- Line 118, but also a general remark to the authors findings: the fact that morbilliviruses can often utilize SLAMs from non-native hosts with similar efficiency but are still quite host-restricted, actually suggests that receptor usage might not be the most important host determinant to me. The authors discuss this in lines 386-407, but in a one-sided manner. I would argue that the fact that morbilliviruses rarely establish themselves in a new host species is an argument that these viruses are extremely host-restricted. Can the authors extend this section of the discussion?

- The data presented in figure 1, 2, and 3 largely show the same results: syncytia in SLAM expressing cells, plaques in SLAM expressing cells, and fusion with SLAM (without HA tag) expressing cells. The authors could condense these data. It is confusing that cow and sheep SLAM were added to figure 3, as they could also have been included in figure 1 and 2.

- What does the fusion assay quantitatively assess? MV seems more fusogenic than other viruses, in conjunction with more efficient plaque formation than other morbilliviruses. What does this mean?

- In line 216-231 the authors extensively explain the interaction between MV-H and HAtag-batSLAM. Why is this relevant, and is it not sufficient to present the findings on the interactions between MV-H and untagged batSLAM?

Reviewer #2: My major criticism to the study is the lack of context of prerequisites for cross-species morbillivirus infection. First, the heterologous target host (or host population) needs to lack morbillivirus-specific immunity. It has been well described that morbillivirus-specific cellular immune responses provide cross-protection to other morbilliviruses. Second, the host needs to be susceptible. I agree with the authors that nectin-4 is much more conserved than CD150, which will likely make CD150-mediated infection of immune cells the critical hurdle. Next, the host cells need to be permissive, and the antiviral proteins need to effectively suppress innate immune responses in the novel host. Finally, and most importantly, the virus needs to be transmitted to the next host. Without onward transmission, all adaptive mutations are lost in the first infected host. All these factors are well-explained in the COV review of 2020 but are lost in the current manuscript.

The abstract is more a narrative than that it summarizes the work and main findings of this manuscript. As far as I can assess, lines 46-61 does not contain any new information. The adaptation experiment of MV to use myotis bat CD150 if new, but the authors do not explain that this is the inverse of what would be a more interesting experiment (although probably not possible due to concerns about gain of function): how easy is it for the bat virus to adapt to use human CD150? A reverse zoonosis transmission of MV to bats is not a likely event. The hypothetical ancestral CD150 molecules proposed in this manuscript are truly novel, and an original approach tu understanding morbillivirus evolution. However, in my opinion the manuscript whould be rewritten to explain the specific aim, the findings and their novelty, and the advance that the study presents to the existing literature.

Reviewer #3: The authors need to carry out immunofluorescense staining on the infected cell lines to validate their conclusions.

**Part III – Minor Issues: Editorial and Data Presentation Modifications**

Reviewer #1: Minor comments

- English could be improved throughout:

o Line 2: ‘Its’ could be deleted.

o Line 48: what does ‘severe cross-species transmission’ mean?

o Line 88: what does ‘major pathogen’ mean

o Line 160 (and other places): rather use ‘infectious virus’ than ‘live virus’

o And many other examples

- Line 47: Why did the authors highlight MV, CDV and CeMV

- Line 59-60: ‘bat SLAM also functioned as an efficient receptor for multiple morbilliviruses’. Is this background or a finding of this paper? Unclear now.

- Line 91: The authors indicate that morbilliviruses lead to economic losses in livestock industries. This only holds true for PPRV, and this should not be a generalized statement.

- Line 97: novel morbilliviruses were not discovered in bats or swine, rather the presence of a sequence was demonstrated.

- Line 103: no need to refer to a figure in the introduction.

- Line 113: has CDV jumped to primates since 2010?

- Line 191: the N187Y mutation does not become directly clear in supplementary figure 5.

- Line 203-207: no need to extensively repeat the findings of the prior paragraph.

- The extensive descriptions of IFIEs and PIEDA is rather technical and not relevant for a general readership. Could these results be summarized in a table shortening the results section extensively?

- Line 531-533: the phylogenetic tree should be part of the supplemental data.

Reviewer #2: 1. Title: the authors are recommended to use the term CD150. If not, they should use the term SLAMF1.

2. Line 48: the term “severe” refers to disease, not to transmission events.

3. Lines 81-82: add “in vitro”: in vivo cross-species transmission events have been described (and all current morbilliviruses are thought to have evolved after cross-species transmission events and adaptation to novel hosts. However, in vivo these events are still rare, and morbillivirus infections are usually restricted to a well-defined range of host species (as also summarized in the 2020 COV review).

4. Lines 127-146 and Figure 1A: these data are largely identical to Figure 2 in Takeda et al Curr Opin Virol 2020. The only addition is the combination of MBaMV and Vero-batCD150. Figure 1B should be moved to the supplemental data. It mostly shows that the PDV strain used in this study was likely somewhat attenuated.

5. Lines 147 and Figure 2: it is unclear to me what the added value is of the plaque assay over the syncytium assay shown in Figure 1. It is unclear why MV did not infect Vero-batCD150 in the first study but still caused small plaques in the plaque assay. This is not explained by the authors.

6. Lines 174-178: have the authors considered testing alternative H and F sequences in the DSP assay?

7. Lines 179-181: the rationale of this adaptation experiment is not well-explained. The results of the experiment are convincing, and well-presented.

8. Lines 203-241: the computational analysis presented in Fig 5 is convincing, but the importance is not well-explained.

9. Lines 303-308: the authors observed “reduced receptor functionality” in cells expressing macaque slam wit Arg28 but conclude that “human SLAM cannot serve as a CDV receptor due to the presence of R28”. The reduced receptor functionality in Fig 6E is substantially higher than the negative control, how can the authors exclude that the virus can adapt? In fact, this has previously been described in references 44 and 45 (see also discussion, lines 383-385).

10. Lines 324-326: the exclusion of bat CD150 in this analysis is unfortunate, as bats are considered potential hosts of ancestral morbilliviruses (see also lines 357-358). This point should be addressed in the discussion.

11. Lines 348-352: CDV is considered as a more recently diverged morbillivirus than MV (Uhl et al 2019, PMID: 30743216). Does this fit the observed picture?

12. Lines 386-397: this paragraph should address cross-protective immunity.

13. Lines 398-399: only if combined with transmissibility to the next host, see comment 1. As transmission happens relatively late in the viral life cycle, there are several other restrictions. Receptor adaptation is necessary but not sufficient.

14. Lines 408-411: I agree with the statement but disagree with the conclusion that this study provides essential scientific evidence for the importance of CD150-adaptation as the basis of cross-species transmission.

Reviewer #3: (No Response)

PLOS authors have the option to publish the peer review history of their article (what does this mean? ). If published, this will include your full peer review and any attached files.

**Do you want your identity to be public for this peer review?** For information about this choice, including consent withdrawal, please see our Privacy Policy .

Reviewer #1: No

Reviewer #2: No

Reviewer #3: No

**Figure resubmission:**
---

## [Decision Letter · Decision Letter 1]

PPATHOGENS-D-25-00434R1

Evolutionary and structural basis of SLAMF1 utilization in morbilliviruses – Implications for host range and cross-species transmission

PLOS Pathogens

Dear Dr. Takeda,

Thank you for submitting your manuscript to PLOS Pathogens. After careful consideration, we feel that it has merit but does not fully meet PLOS Pathogens's publication criteria as it currently stands. Therefore, we invite you to submit a revised version of the manuscript that addresses the points raised during the review process.

Please submit your revised manuscript within 30 days Jul 07 2025 11:59PM. If you will need more time than this to complete your revisions, please reply to this message or contact the journal office at plospathogens@plos.org. Please include the following items when submitting your revised manuscript:

We look forward to receiving your revised manuscript.

Kind regards,

Barry Rockx

Academic Editor

PLOS Pathogens

Thomas Hoenen

Section Editor

PLOS Pathogens

Sumita Bhaduri-McIntosh

Editor-in-Chief

PLOS Pathogens

orcid.org/0000-0003-2946-9497

Michael Malim

Editor-in-Chief

PLOS Pathogens

orcid.org/0000-0002-7699-2064

**Additional Editor Comments :**

Thank you for addressing the reviewer comments in the revise manuscript. As you can see below, there are still a few few minor comments that require your attention.

**Journal Requirements:**

1) We have noticed that you have uploaded Supporting Information files, but you have not included a complete list of legends. Please add a full list of legends for your Supporting Information file ( Data set.xlsx ) after the references list. 2) Please amend your detailed Financial Disclosure statement. This is published with the article. It must therefore be completed in full sentences and contain the exact wording you wish to be published.1) State the initials, alongside each funding source, of each author to receive each grant. For example: "This work was supported by the National Institutes of Health (####### to AM; ###### to CJ) and the National Science Foundation (###### to AM).". 3) Thank you for stating that "Animal silhouette images were generated using OpenAI’s image generation system (DALL·E)." Please ensure to include a direct link to the Terms and Conditions of the AI software in the figures legends. 

**Reviewers' Comments:**

Reviewer's Responses to Questions

**Part I - Summary**

Reviewer #1: In this revised version of the manuscript, Hyodo et al addressed all my comments satisfactory. I have some final comments that could be adapted to improve the manuscript.

Reviewer #3: The authors have made major revisions to the manuscript, including adding the requested immunofluorescence experiments. The manuscript is therefore greatly improved. However, a remaining point that needs addressed is that on page 8 relating to supplementary figure 5, there is some infection at 18 hours in the parental Vero cells, indicating that another receptor rather than SLAM may be used. In fact, it has been shown in other studies that if you leave the infection longer it will spread throughout the monolayer and give good virus titres. Therefore while SLAM usage certainly gives better fusion and faster virus spread SLAM may not be necessary for infection of these viruses. This point needs addressed.

**Part II – Major Issues: Key Experiments Required for Acceptance**

Reviewer #1: None

Reviewer #3: (No Response)

**Part III – Minor Issues: Editorial and Data Presentation Modifications**

Reviewer #1: • The abstract has improved majorly, but I find it quite lengthy. I suggest the authors condense line 58-89 to highlight their most important novel findings.

• The authors call the morbillivirus evolution a ‘recent example’ (line 510-511). I guess time is relative, but calling 600 BC recent might not be appropriate.

• Line 518-519: I find it not strong to end this section with a question, and suggest the authors revise that sentence accordingly.

Reviewer #3: (No Response)

PLOS authors have the option to publish the peer review history of their article (what does this mean? ). If published, this will include your full peer review and any attached files.

**Do you want your identity to be public for this peer review?** For information about this choice, including consent withdrawal, please see our Privacy Policy .

Reviewer #1: No

Reviewer #3: No

**Figure resubmission:**
---

## [Editor Report · Decision Letter 2]

Dear Dr. Takeda,

We are pleased to inform you that your manuscript 'Evolutionary and structural basis of SLAMF1 utilization in morbilliviruses – Implications for host range and cross-species transmission' has been provisionally accepted for publication in PLOS Pathogens.

Best regards,

Barry Rockx

Academic Editor

PLOS Pathogens

Thomas Hoenen

Section Editor

PLOS Pathogens

Sumita Bhaduri-McIntosh

Editor-in-Chief

PLOS Pathogens

orcid.org/0000-0003-2946-9497

Michael Malim

Editor-in-Chief

PLOS Pathogens

orcid.org/0000-0002-7699-2064

all reviewer comments are adequately addressed.
---

## [Editor Report · Acceptance letter]

Dear Dr. Takeda,

We are delighted to inform you that your manuscript, "Evolutionary and structural basis of SLAMF1 utilization in morbilliviruses – Implications for host range and cross-species transmission," has been formally accepted for publication in PLOS Pathogens.

Best regards,

Sumita Bhaduri-McIntosh

Editor-in-Chief

PLOS Pathogens

orcid.org/0000-0003-2946-9497

Michael Malim

Editor-in-Chief

PLOS Pathogens

orcid.org/0000-0002-7699-2064